# Learning Feasibility to Imitate Demonstrators with Different Dynamics

**Zhangjie Cao[1], Yilun Hao[1], Mengxi Li[2], Dorsa Sadigh[1,2]**
[1]Department of Computer Sciences, Stanford University, United States
[2]Department of Electrical Engineering, Stanford University, United States
`caozj@cs.stanford.edu, {yilunhao,mengxili}@stanford.edu, dorsa@cs.stanford.edu`

**Abstract:** The goal of learning from demonstrations is to learn a policy for an agent (imitator) by mimicking the behavior in the demonstrations. Prior works on learning from demonstrations assume that the demonstrations are collected by a demonstrator that has the same dynamics as the imitator. However, in many real-world applications, this assumption is limiting — to improve the problem of lack of data in robotics, we would like to be able to leverage demonstrations collected from agents with different dynamics. This can be challenging as the demonstrations might not even be feasible for the imitator. Our insight is that we can learn a feasibility metric that captures the likelihood of a demonstration being feasible by the imitator. We develop a feasibility MDP (f-MDP) and derive the feasibility score by learning an optimal policy in the f-MDP. Our proposed feasibility measure encourages the imitator to learn from more informative demonstrations, and disregard the far from feasible demonstrations. Our experiments on four simulated environments and on a real robot show that the policy learned with our approach achieves a higher expected return than prior works. We show the videos of the real robot arm experiments on our website.

**Keywords:** Imitation Learning, Learning from Agents with Different Dynamics

## 1 Introduction

Imitation learning aims to learn a well-performing policy from demonstrations. Standard imitation learning algorithms usually assume that the *demonstrator* (the agent that generates the demonstrations) and the *imitator* (the agent that is learning a policy) share the same dynamics, i.e., the transition functions are the same [1, 2, 3, 4]. Specifically, in a given state, with the same action, both the demonstrator and the imitator transition to the same distribution of next states. However, this assumption limits the usage of already collected demonstrations. Imagine a setting, where a set of demonstrations are collected on a 7 Degrees of Freedom (DoF) robot arm shown in Fig. 1 to place a book on the empty area of the shelf (on the left) without colliding with the books that are already placed on the right side of the shelf. Later, we might decide to buy a different arm with 3 DoF (e.g., only the joints circled in green as shown in the figure are used). We would like to learn a policy for this 3 DoF robot arm that can achieve the same task—placing the book on the empty region of the shelf—using the originally collected demonstrations on the 7 DoF arm. In general, our goal is to enable using and reusing data collected on robots with different dynamics or embodiments to tackle the problem of lack of in-domain data in robotics. The 3 DoF robot arm should still be able to learn a policy based on feasible or nearly feasible demonstrations from an agent with different dynamics, e.g., using the trajectories that go over the bookshelf in Fig. 1. Motivated by this example, we relax the assumption of shared dynamics between the imitator and demonstrator so that the data can be collected from demonstrators with the same state space but different dynamics from the imitator, e.g., demonstrators with different embodiments, body schemas, joints, or rigid body structures.

Prior works in imitation learning from demonstrators with different dynamics typically rely on state-only demonstrations and learn a policy to maximally follow the sequence of states in demonstrations [5, 6]. Such learning techniques assume that all of the collected demonstrations are useful for the imitator. However, it is possible that demonstrations drawn from agents with other dynamics can

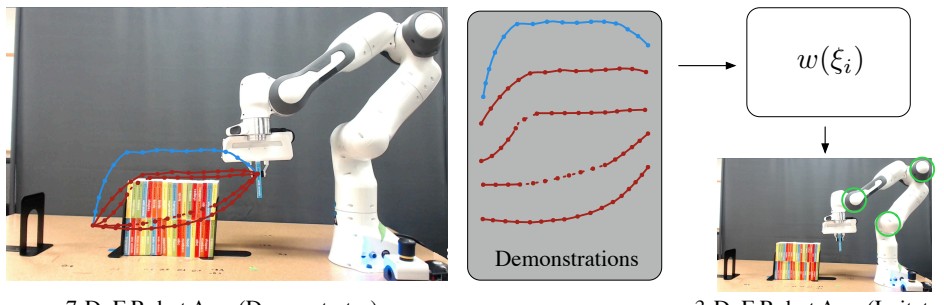

| 7-DoF Robot Arm (Demonstrator) | Demonstrations | 3-DoF Robot Arm (Imitator) |

Figure 1: An example of imitating demonstrators with feasibility. The left image shows that a set of demonstrations (blue and red trajectories) are available for the 7 DoF robot arm. We aim to learn a policy for the 3 DoF robot (joints are circled in green) by learning from the demonstrations of the 7 DoF robot (blue is feasible and red is infeasible). We learn a feasibility score to reweight each demonstration to conduct imitation learning.

be useless or even harmful for the imitator because they may not be feasible for the imitator. Going back to the example in Fig. 1, the red trajectories that move around the stack of books are not feasible for the 3 DoF robot arm. Imitating such trajectories may cause the 3 DoF robot arm to maximally follow these trajectories and even collide with the existing stack of books. Therefore, it is crucial to identify and avoid trajectories that are far from feasible for the imitator, and instead learn more from useful demonstrations, e.g., the blue trajectories that go over the shelf that are still feasible for a robot with 3 DoF.

To avoid the influence of useless or harmful demonstrations from agents with different dynamics, we rely on a *feasibility score*, which measures how feasible a trajectory is for the imitator, and select trajectories with high feasibility to imitate. For example, the blue trajectories should have higher feasibility than the red trajectories in Fig. 1. Prior work such as Cao and Sadigh [7] estimate the feasibility score by computing the distances of demonstrations and corresponding trajectories but the performance highly relies on the accuracy of the inverse dynamics model, which can be difficult to learn. Our key idea is to directly learn a feasibility score for the imitator based on the collected demonstrations. Specifically, we model the imitator environment as an MDP and build a feasibility Markov Decision Process (f-MDP) based on the imitator's MDP and the trajectories provided by the demonstrator. The optimal policy for the f-MDP maximally follows the behavior of the demonstrations but is limited by the imitator's environment. This optimal policy helps assign a feasibility score over the demonstrations. We conduct imitation learning on the demonstrations re-weighted by the feasibility score to learn the final policy for the imitator. We experiment with several simulation environments and a manipulation task with a Panda Franka arm. We show that the policy learned from demonstrations re-weighted by our feasibility achieves higher performance compared to other methods.

## 2   Related Works

**Imitation Learning.** Imitation learning seeks a policy that best imitates demonstrations. Current imitation learning methods can be roughly divided into Behavior Cloning (BC), Inverse Reinforcement Learning (IRL) and Generative Adversarial Imitation Learning (GAIL). BC directly learns the policy from a sequence of state-action pairs via supervised learning [8], where dataset aggregation [9] or policy aggregation [10, 11] are proposed to address the compounding errors problem. IRL first learns a reward function that best matches demonstrations and then finds a policy through reinforcement learning to maximize the recovered reward [1, 12, 2, 13]. GAIL learns the expert policy by matching the occupancy measure between the policy and the demonstrations [4].

However, most imitation learning works require that the demonstrations consist of a sequence of states and actions. When only state observations are available, new imitation learning algorithms are proposed to address the lack of actions. Torabi et al. [14] recover the actions between consecutive states through an inverse dynamics model. GAIL-based works directly match the state occupancy measure between the demonstrations and the policy [15, 16, 17]. However, imitation learning methods learned from either state-action or state-only demonstrations assume that the demonstrator and the

imitator have the same dynamics. Since demonstrations from different dynamics may not be feasible for the imitator, directly imitating cannot achieve the same optimal behavior, and may cause unknown suboptimal outcomes. Thus, standard imitation learning algorithms do not fit our problem setting.

**Learning from Demonstrations with Different Dynamics.** Early works model this problem as a correspondence problem between the demonstrator and the imitator, and map states and actions in demonstrations to the imitator's states and actions [18, 19]. Englert et al. [20] align the state trajectory distributions to address the correspondence problem. Calinon et al. [21] model the demonstrations as a Gaussian mixture model within a projected lower-dimensional subspace. Eppner et al. [22] learn a task description. Domain randomization methods learn the correspondence as an invariant latent space by randomizing domains [23, 24, 25]. Zhang et al. [26] learns a translation mapping to model the correspondence. However, modeling correspondence requires that there exists a strict correspondence between the MDP of the demonstrator and the imitator. Recent works instead only assume the shared state space between the demonstrator and the imitator, and address the different dynamics problem by encouraging the imitator to maximally follow the state trajectory of the demonstrator [5, 27, 28, 6]. However, all these works ignore an important challenge—that is the demonstrations may be far from feasible for the imitator. Enforcing the imitator to follow such trajectories may lead to unknown behavior. We focus on this challenge and develop a feasibility score to down-weight demonstration trajectories that are far from feasible for the imitator. Compared to the works that learn feasibility to filter infeasible trajectories [7], we do not require the inverse dynamics model, which can make our setting more generalizable to different environments.

## 3 Problem Statement

In our problem setting, an imitator aims to learn from demonstrations collected from $N$ demonstrators with various dynamics. We formalize the demonstrators and the imitator each as a standard Markov decision process (MDP). For each demonstrator $j$, ($1 \leq j \leq N$), the MDP is formalized as $\mathcal{M}_j^d = \langle \mathcal{S}, \mathcal{A}_j^d, p_j^d, \mathcal{R}, \rho_0, \gamma \rangle$. The MDP for the imitator is $\mathcal{M}^i = \langle \mathcal{S}, \mathcal{A}^i, p^i, \mathcal{R}, \rho_0, \gamma \rangle$. $\mathcal{S}$ is the shared state space for all environments. $\mathcal{A}_j^d$ and $\mathcal{A}^i$ are the action spaces and $p_j^d : \mathcal{S} \times \mathcal{A}_j^d \times \mathcal{S} \to [0, 1]$ and $p^i : \mathcal{S} \times \mathcal{A}^i \times \mathcal{S} \to [0, 1]$ are the transition probabilities for each demonstrator and the imitator respectively. Note that in our problem setting, we use the transition function $p$ to denote dynamics and the demonstrators and the imitator may have different dynamics and action spaces. $\rho_0$ is the shared initial state distribution for all MDPs. $\mathcal{R} : \mathcal{S} \times \mathcal{S} \to \mathbb{R}$ is the reward function. Note that we make the assumption that the reward function is based on state transitions and is shared between the demonstrators and the imitator, which is a common assumption used in prior work [5, 7], and is usually satisfied since the demonstrators and the imitator conduct the same task in the same context. $\gamma$ is the shared discount factor. A policy for the imitator $\pi^i : \mathcal{S} \times \mathcal{A}^i \to [0, 1]$ defines a probability distribution over the space of actions in a given state. An optimal policy $\pi^*$ maximizes the expected return $\eta_{\pi^i} = \mathbb{E}_{s_0 \sim \rho_0, \pi^i} \left[ \sum_{t=0}^{\infty} \gamma^t \mathcal{R}(s_t, a_t, s_{t+1}) \right]$, where $t$ indicates the time step.

We aim to learn a policy $\pi^i$ for the imitator, given a set of demonstrations from different demonstrators $\Xi^j = \{\xi_1^j, \ldots, \xi_D^j\}_{j \in \{1 \ldots N\}}$ where each trajectory is a sequence of states $\xi = \{s_0^d, s_1^d, \ldots, s_H^d\}$. We assume that *the optimal policy can be learned by imitating the useful demonstrations*, which is a general assumption adopted by prior imitation learning works [8, 4, 5, 7]. The violation of this assumption, as shown in prior works, leads to learning a suboptimal policy. Note that *we discard actions from the demonstrations* instead of imitating the state-action trajectories because different action spaces between the demonstrators and the imitator make it impossible to imitate the actions.

**Challenges.** The core challenges of imitation learning from demonstrations with different dynamics are: (1) How to imitate useful demonstrations with different dynamics, (2) How to avoid harmful demonstrations misleading the imitator. Prior works have studied and made progress for the first challenge [5, 6], but the second challenge is still under-studied. Strong assumptions such as access to or learning an accurate inverse dynamics model are needed to filter out harmful demonstrations [7]. We address the second challenge by learning a feasibility score that measures how likely it is for a demonstration to be feasible for the imitator with minimal assumptions: only using the environment of the imitator, i.e., we can collect interaction data in the environment but we do not know the exact reward and transition function of the imitator.

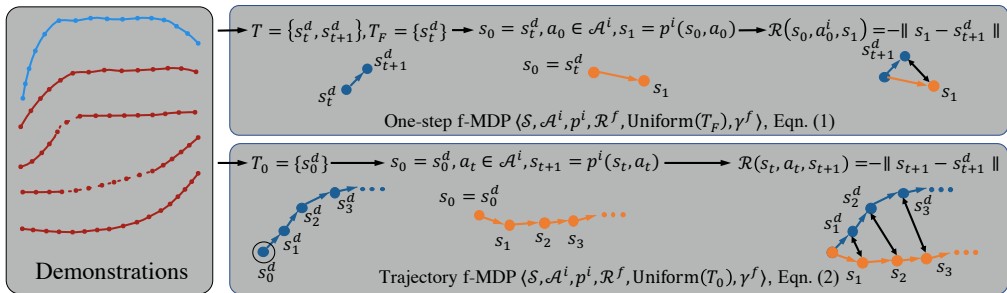

Figure 2: The illustration and comparison of one-step f-MDP and trajectory f-MDP. The blue state transition and trajectory are from the demonstrations while the orange state transition and trajectory are rollouts in the f-MDP. One-step f-MDP collects the states in the *Former Set* and uses the uniform distribution over the states as the initial state distribution. Trajectory f-MDP collects the initial state of all the demonstrations and uses a uniform distribution over them as the initial state distribution.

## 4 Feasibility-Based Imitation Learning

The feasibility of a trajectory depends on the feasibility of each state transition in the trajectory, i.e., if $(s_t, s_{t+1})$ is feasible for all time steps. A state transition $(s_t, s_{t+1})$ is feasible when there exists an action $a_t^i \in A^i$ such that $p^i(s_t, a_t^i, s_{t+1}) = 1$ for deterministic transitions or $p^i(s_t, a_t^i, s_{t+1}) > 0$ for stochastic transitions. In this section, we discuss the deterministic MDP setting and discuss the stochastic setting in Appendix.

Feasibility can be directly measured by a perfect inverse dynamics model $f : \mathcal{S} \times \mathcal{S} \to \mathcal{A}$ that takes a state transition $(s_t, s_{t+1}) \in \mathcal{S} \times \mathcal{S}$ as the input and outputs the action $a_t \in \mathcal{A}$ that achieves the transition if feasible or outputs 'Infeasible'. However, having access to this model is often non-trivial and such a binary feasibility measurement as $f$ discards all infeasible demonstrations without considering any useful information from slightly infeasible trajectories.

Our goal is to learn a policy $\pi^i : \mathcal{S} \to \mathcal{A}^i$ for the imitator to maximally achieve the state transitions in the demonstrations. This means that if the state transition $(s_t^d, s_{t+1}^d)$ from a demonstration is feasible, the next state produced by $\pi^i$, i.e., $s_{t+1}^i = p^i(s_t^d, \pi^i(s_t^d))$ should be equal to $s_{t+1}^d$. Otherwise, we would like the policy to output an action that ensures the next state $s_{t+1}^i$ is as close as possible to the next state from the demonstration $s_{t+1}^d$. Therefore, the distance between $s_{t+1}^i$ and $s_{t+1}^d$ can serve as a measure of feasibility, where a smaller distance corresponds to a higher likelihood of feasibility. To learn this policy, we design a feasibility MDP (f-MDP), where we ensure that the optimal policy of the f-MDP satisfies the above requirement. f-MDP is defined as $M^f = \langle \mathcal{S}, \mathcal{A}^i, p^i, \mathcal{R}^f, \rho_0^f, \gamma^f \rangle$. We will now discuss our choices for the components of f-MDP.

**One-step f-MDP.** First, recall that our goal is to learn a policy *for the imitator* to *maximally achieve the state transitions in the demonstrations*. So the policy should be learned in an environment with the same state-action space and transition probability as the imitator. We would like the reward of the f-MDP to encourage maximally achieving the state transitions in the demonstrations. Let us first collect all the state transitions $T = \{(s_t^d, s_{t+1}^d)\}$ in all of the demonstrations. We define the *Former Set* to be the set of states in the demonstrations that one can transition from: $T_F = \{s_t^d : (s_t^d, s_{t+1}^d) \in T\}$. The initial state distribution $\rho_0^f$ can be defined uniformly over the Former Set as $\text{Uniform}(T_F)$. Here, we assume that all the states in the Former Set can be visited by the imitator. We define the reward of a *One-step f-MDP* so that it matches the one-step transitions from the Former Set:

$$s_t^d \sim \text{Uniform}(T_F), \quad s = s_t^d, \quad s' = p^i(s, a), \quad R^f(s, a, s') = -f_{\text{dis}}(s', s_{t+1}^d), \qquad (1)$$

where $(s_t^d, s_{t+1}^d)$ is a state transition in the demonstrations and $a \in A^i$ is sampled from the action space of the f-MDP. $f_{\text{dis}}$ is a function that measures the distance between the states (e.g., the L2 distance). We define the reward to penalize the distance between $s'$ and $s_{t+1}^d$.

**Trajectory f-MDP.** The one-step f-MDP suffers from an important shortcoming: the assumption that all states in the Former Set must be visited by the imitator can be violated, because the demonstrators have different dynamics from the imitator and some demonstration states can never be reached by the imitator. So we cannot set $\text{Uniform}(T_F)$ as the initial state distribution for the f-MDP. We instead

collect the initial state $s_0^d$ of all the demonstrations, $T_0 = \{s_0^d\}$, and define the initial state distribution of the *Trajectory f-MDP* as Uniform($T_0$). Since all the demonstrators and the imitator share the initial state distribution, all states in $T_0$ can be visited by the imitator. We define the reward as:

$$s_0^d \sim \text{Uniform}(T_0), \quad s_0 = s_0^d, \quad s_{t+1} = p^i(s_t, a), \quad R^f(s_t, a, s_{t+1}) = -f_{\text{dis}}(s_{t+1}, s_{t+1}^d), \quad (2)$$

We use the L2 distance for $f_{\text{dis}}$ in our experiments. Similar to the one-step f-MDP $a \in A^i$ is sampled from the action space of the imitator.

**Learning Feasibility.** Given the Trajectory f-MDP defined above, for each demonstration trajectory $\xi$, the highest reward achieved in this f-MDP reflects the feasibility score of the trajectory. We use reinforcement learning to learn the optimal policy of the Trajectory f-MDP, $\pi^*$. We then derive the feasibility of each demonstration trajectory $\xi$ as a function of the trajectory f-MDP reward:

$$w(\xi) = \exp\left(\frac{-\sum_{t=1}^N (\gamma^f)^t f_{\text{dis}}(s_t, s_t^d) - C}{\sigma}\right). \quad (3)$$

$s_t$ is the state at step $t$ in the rollout derived by the policy $\pi^*$. We use an exponential function of the cumulative reward since the cumulative reward is always negative and the exponential function can bound the feasibility in the range of $[0, 1]$. The parameter $C$ is used to shift the function to avoid the situation where the cumulative reward is extremely negative, while the parameter $\sigma$ controls how low the reward can be, and when a demonstration can be fully filtered out by assigning a feasibility of close to $0$. In practice, $C$ is usually set as the maximal cumulative reward over all demonstrations to ensure the maximal feasibility is $1$.

For the feasibility of each state transition $(s_t^d, s_{t+1}^d)$, we use the feasibility of the trajectory it belongs to: $w((s_t^d, s_{t+1}^d)) = w(\xi_i)$, where $(s_t^d, s_{t+1}^d) \in \xi_i$. We do not use the state distance at each time step between $s_{t+1}$ and $s_{t+1}^d$ as in the One-step f-MDP because such measurement suffers from the fact that within a trajectory, the reward of later steps are influenced by former steps. For example, if $s_t$ diverges from $s_t^d$, $s_{t+1}$ will diverge more from $s_{t+1}^d$. So the per-step reward is an unfair measure of feasibility for the state transition $(s_t^d, s_{t+1}^d)$ at different time steps $t$. Therefore, we use the accumulative reward of the whole trajectory as our feasibility measure, where all the state transitions share the same value.

The discount factor $\gamma^f$ is usually set as $\gamma^f < 1$ to reduce compounding errors. Specifically, the length of a rollout in the f-MDP is the same as the corresponding demonstration, which can be very long. If the state in the rollout starts to diverge from the demonstration trajectory at $t$, meaning that $\|s_t - s_t^d\| > 0$, the steps after time step $t$ even diverge more from the demonstration. This makes the trajectory reward for all the infeasible trajectories very low and does not allow for discriminating among different infeasible trajectories. Therefore, we set a discount factor of $\gamma^f < 1$ to discount or even ignore the trajectory reward at later steps.

Leveraging our Trajectory f-MDP design, feasible trajectories still receive the maximal reward of $0$ since each state in the rollout will perfectly match the demonstration thus having a feasibility of $1$ as in Eqn. (3). Instead, infeasible trajectories receive negative rewards leading to smaller feasibility scores, which reflects how far away the demonstration is from the closest feasible trajectory.

One may worry about the time complexity of our approach since we need an additional RL training to learn an optimal policy for the f-MDP. However, the f-MDP is a lot simpler compared to the imitator's MDP since the initial distribution is reduced from the distribution of all possible states in the demonstration set to a discrete distribution over the initial states of the demonstrations. This can simplify the time complexity of finding the optimal policy for the f-MDPs.

**Algorithm.** Using the feasibility metric in Eqn. (3), we assign each state transition with the same feasibility of the trajectory it belongs to. Directly weighing the imitation loss as [29] may lead to gradients that are close to $0$ if a batch of data all have low feasibility. This can make the algorithm inefficient by wasting samples from many iterations. Instead, for a more efficient training, we define a discrete probability distribution $p_w$ over the collection of state transitions in all the demonstrations: $T$, where the probability of a state-transition $(s_t^d, s_{t+1}^d)$ as $p_w((s_t^d, s_{t+1}^d)) = \frac{w((s_t^d, s_{t+1}^d))}{\sum_{\left(s_{t'}^d, s_{t'+1}^d\right) \in T} w((s_{t'}^d, s_{t'+1}^d))}$.

State transitions with larger feasibility will be sampled more often. Using the sampling distribution $p_w$, we can embed our method into any imitation learning algorithm to enable learning from demonstrations with different dynamics. We show the algorithm block in the Appendix.

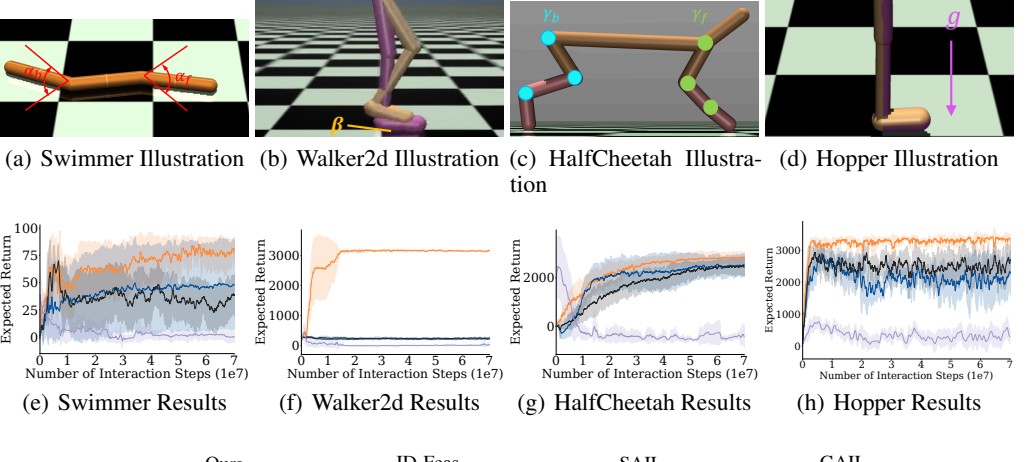

(a) Swimmer Illustration  (b) Walker2d Illustration  (c) HalfCheetah Illustration  (d) Hopper Illustration

(e) Swimmer Results  (f) Walker2d Results  (g) HalfCheetah Results  (h) Hopper Results

Ours — ID-Feas — SAIL — GAIL

Figure 3: Illustration of different dynamics in (a) Swimmer: varying the joint limit of the front and back joints ($\alpha_f$ and $\alpha_b$). (b) Walker2d: varying the friction of the feet ($\beta$). (c) HalfCheetah: varying the joint control force of the front and back joints by multiplying a factor $\gamma_f$ and $\gamma_b$ with the front and back joint force. (d) Hopper: varying the gravitational constant respectively (e-h) show the expected return in these four environments.

**Sampling More Demonstrations with the Feasibility Score.** When the existing useful demonstrations are too scarce to learn a well-performing imitation learning policy, we need to acquire more demonstrations from the demonstrators. But collecting new demonstrations can be expensive, so we often can only acquire a limited budget of demonstrations. We thus need to collect the most useful demonstrations within this limited budget. The proposed feasibility metric provides a criterion to decide the similarity between the imitator and each demonstrator. If a demonstrator has a higher similarity, we sample more from this demonstrator because its demonstrations are more likely to be feasible. Specifically, we create a probability distribution $p_j$ over all demonstrators:

$$p_j = \frac{\frac{1}{|\Xi^j|} \sum_{\xi^j \in \Xi^j} w(\xi^j)}{\sum_{j=1}^{N} \frac{1}{|\Xi^j|} \sum_{\xi^j \in \Xi^j} w(\xi^j)}. \tag{4}$$

We repeatedly and independently query the demonstrator $j$ according to $p_j$ and collect a demonstration. The proposed sampling strategy samples more demonstrations from closer demonstrators. We empirically show that the sampling strategy derived from our feasibility performs better than uniform sampling or using other feasibility metrics as in [7].

## 5 Experimental Results

We experiment with four MuJoCo environments, a simulated Franka Panda Arm, and a real Franka Panda Arm. We also show results on various compositions of demonstrations of different dynamics and the performance gain when we are given a larger budget to collect demonstrations. We compare our approach with a standard imitation learning algorithm: GAIL [4], imitation learning from demonstrations with different dynamics methods without a measure of feasibility: SAIL [5], and with a feasibility score: ID-Feas [7], which uses an inverse dynamics model to estimate feasibility.

### 5.1 MuJoCo Experiments

**Swimmer.** The swimmer agent has three links and two joints. The goal of the agent is to move forward by rotating the joints. As shown in Fig. 3(a), we create different dynamics by setting the joint limit of the front and the back joints, denoted by $(\alpha_f, \alpha_b)$. The original Swimmer environment has $(\alpha_f, \alpha_b) = (100°, 100°)$. We create four demonstrator environments $(\alpha_f, \alpha_b)$: (i) $(100°, 12°)$, (ii) $(100°, 20°)$, (iii) $(100°, 100°)$, and (iv) $(10°, 100°)$. We also create the imitator environment by setting $(\alpha_f, \alpha_b) = (100°, 10°)$. The demonstrators (i) and (ii) are closer to the imitator in terms of their dynamics, while the demonstrators (iii) and (iv) are farther.

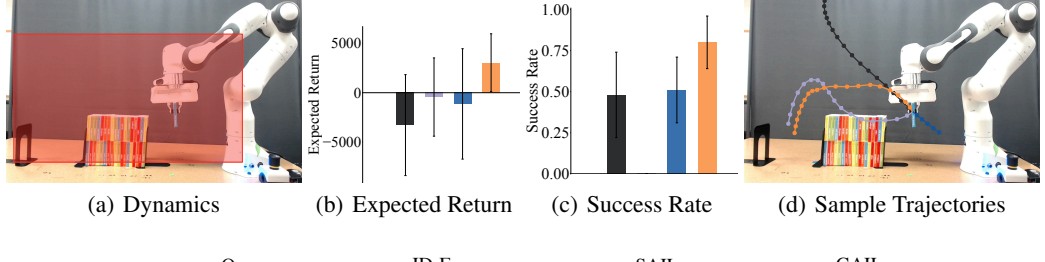

|  (a) Dynamics  |  (b) Expected Return  |  (c) Success Rate  |  (d) Sample Trajectories  |

Ours — ID-Feas — SAIL — GAIL

Figure 4: (a) Illustration of different dynamics in the real robot arm environment. (b-c) The bar plots show the expected return and success rate compared to other methods. (d) Sampled trajectories using different methods.

**Walker2d.** The Walker2d is an agent with two legs where each leg consists of 3 joints. We create different dynamics by using different frictions $\beta$ for the feet, i.e., the link that touches the ground. The original Walker2d uses $\beta = 0.9$. We create two settings to show high friction and low friction of the imitator with a mix of frictions for the demonstrators. In the first setting, there are four demonstrators: (i) $\beta = 19.9$, (ii) $\beta = 9.9$, (iii) $\beta = 0.9$, and (iv) $\beta = 0.7$. The imitator has $\beta = 24.9$. In the second setting, there are four demonstrators: (i) $\beta = 29.9$, (ii) $\beta = 19.9$, (iii) $\beta = 1.1$, and (iv) $\beta = 0.7$. The imitator has $\beta = 0.9$.

**HalfCheetah.** The HalfCheetah is an agent with two legs at the front and back of the body, where each leg consists of three joints. We create different dynamics by varying the control force limit of joints of the front leg and back leg, where we multiply a factor $\gamma_f$ with the original control force limit of the front leg and multiply $\gamma_b$ with the limit of the back leg. We create two settings, where the demonstrators have low and high similarity with each other. In the first setting, there are four demonstrators with $(\gamma_f, \gamma_b)$: (i) $(0.05, 1)$, (ii) $(0.5, 1)$, (iii) $(1, 0.5)$, and (iv) $(1, 0.05)$. The imitator has $(\gamma_f, \gamma_b) = (0.01, 1)$. In the second setting, there are four demonstrators with $(\gamma_f, \gamma_b)$: (i) $(0.01, 1)$, (ii) $(0.05, 1)$, (iii) $(1, 0.05)$, and (iv) $(1, 0.01)$. The imitator has $(\gamma_f, \gamma_b) = (0.01, 1)$.

**Hopper.** The Hopper is an agent with one leg consisting of 3 joints. We create different dynamics by using different gravitational constants $g$. The original Hopper uses $g = 9.81$. We create four demonstrator environments: (i) $g = 20.0$, (ii) $g = 9.81$, (iii) $g = 5.0$, and (iv) $g = 2.0$. We also create the imitator environment by setting $g = 15.0$.

The detailed composition of demonstrations for all four environments is included in the Appendix. For all the Mujoco environments, we evaluate the expected return of each policy by rolling out 100 trajectories in the environment with the policy and compute the average expected return of the 100 trajectories. We run each experiment for 5 times and show the mean and the standard deviation.

**Results.** We show the expected return with respect to the number of steps for the four different environments in Fig. 3. We show the results of the second setting for the Walker2d and the HalfCheetah in the Appendix. We observe that our proposed feasibility achieves the best performance among all the methods. The highest p-value comparing our method to baselines is $0.116$ with ID-Feas for Swimmer, $2.55e - 14$ with GAIL for Walker2d, $0.188$ with ID-Feas for HalfCheetah, and $0.026$ with GAIL for Hopper. In particular, our method outperforms ID-Feas, which indicates that the proposed feasibility more accurately filters out far from feasible demonstrations. SAIL performs even worse than GAIL, this is because SAIL can more strictly follow the state sequences of demonstrations than GAIL including those far from feasible demonstrations. Our demonstration set is composed of a high percentage of demonstrations from unrelated dynamics, which can mislead SAIL's learned policy.

## 5.2 Simulated and Real Robot Arm Experiments

**Setup.** We create a simulated robot arm based on a Panda Robot Arm implemented in the PyBullet [30] and a real robot arm environment using a Franka Panda Arm[1]. We include the results for the simulated robot arm in the Appendix. As shown in Fig. 4(a), we create a task of moving a book to the shelf but the closest region on the shelf is full. So we need to move the book to the empty area of the shelf without colliding with the shelf and the existing books on the shelf. We create two dynamics for the robot arm: using a 7-DoF control which can move freely in the 3D space, and using

---

[1]https://www.franka.de

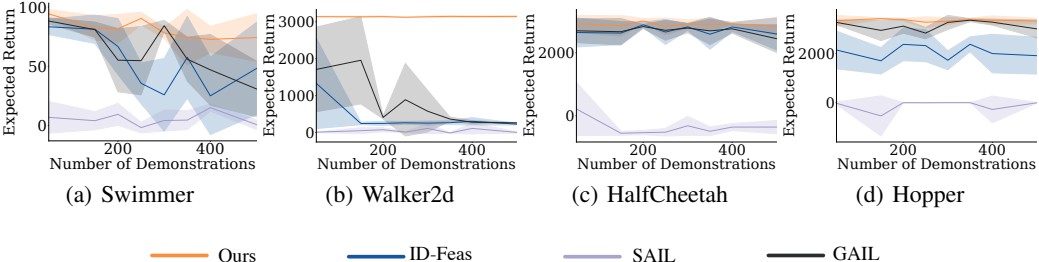

Figure 5: (a-d) The expected return when increasing the number of demonstrations from agents with unrelated dynamics. The results in Fig. 3 correspond to using 500 demonstrations from each unrelated dynamics. In both of these settings, there will also be a fixed number of demonstrations from agents with related dynamics as shown in Appendix.

a 3-DoF control, which is limited to moving on the red plane area. We collect demonstrations from both 7-DoF and 3-DoF controllers and aim to learn an optimal policy for the 3-DoF robot.

For evaluation, we use two metrics: (1) The expected return based on a reward penalizing collision with the shelf and existing books while rewarding the successfully moving the book to the empty area of the shelf within the time limit. More detail on the reward is in the Appendix. (2) The success rate of finishing the task over 100 trials.

We observe that the proposed approach outperforms the baseline methods both in expected return and success rate as shown in Fig. 4. The highest p-value for the expected return is $2.432 \times 10^{-7}$ and for the success rate is $3.534 \times 10^{-8}$ (both with ID-Feas), which are statistically significant.

## 5.3 Analysis

We conduct experiments with varying compositions of demonstrations and investigate the performance of different approaches when we have the budget to acquire additional demonstrations. We show the results of varying the number of demonstrations from all demonstrators in the Appendix.

**Varying the Number of Demonstrations from each Unrelated Demonstrator.** For the first three experiment settings in the Mujoco environment, we have two demonstrators with similar dynamics to the imitator and two demonstrators with far apart dynamics. We vary the number of demonstrations from the far apart demonstrators to investigate their influence on the different methods. We conduct experiments on the first setting for the Swimmer, Walker2D, HalfCheetah, and Hooper and report the results in Fig. 5(a), 5(b), 5(c) and 5(d). With an increasing number of demonstrations from the far apart demonstrators, the expected return of all the methods decreases, while our method shows the best performance consistently across different numbers of demonstrations. This demonstrates that our feasibility can effectively filter out far from feasible demonstrations and ensure the policy learns from useful demonstrations.

## 6 Conclusion

**Summary.** We propose an algorithm to learn a feasibility metric to imitate demonstrations drawn from agents with different dynamics. Our feasibility metric captures how likely it is for each demonstration to be feasible for the imitator. We develop a feasibility MDP (f-MDP) and derive the feasibility by learning the optimal policy for the f-MDP. We show that the policy learned from the demonstrations reweighted by the proposed feasibility score outperforms other imitation learning methods in various environments and different compositions of demonstrations.

**Limitations and Future Work.** Our work only addresses the problem of filtering out far from feasible demonstrations, but does not solve the problem of learning a policy from those feasible but suboptimal or nearly feasible demonstrations from different dynamics. There are situations where demonstrations are feasible but not optimal for the imitator, especially when the ability of the demonstrator is more restricted than the imitator. In the future, we aim to study these more general settings.

**Acknowledgments**

We would like to thank FLI grant RFP2-000, NSF Awards 1941722 and 1849952, and DARPA HiCoN project for their support of this project.

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
