# OpenReview forum: "Learning Feasibility to Imitate Demonstrators with Different Dynamics"
_robot-learning.org/CoRL/2021/Conference — CoRL2021 Poster_

### Official Review · Reviewer_Srqj · 2021-07-20

**Originality:** Very Good
**Technical Quality:** Good
**Clarity Of Presentation:** Very Good
**Impact:** 4

**Recommendation:**

Weak Accept: I recommend accepting the paper, but will not argue for my recommendation if the majority of other reviewers have a different opinion.

**Summary:**

The paper considers the problem of imitation learning with demonstrations from potentially different dynamics. The authors propose a new feasibility score which evaluates how feasible it is to reproduce the demonstration with the imitator's dynamics. The feasibility score is obtained from an f-MDP, which shares the state space, action space, transition function with the imitator, but has a reward function quantifying state deviation from the demonstrations. This feasibility can be further used in a more interactive setting to determine which demonstrator to query new trajectories. The authors show in various tasks that the newly proposed feasibility score can help the learner rule out infeasible demonstrations and hence achieve better task performance.

**Issues:**

- The authors are advised to modify a few statements so that they are true for stochastic MDPs, as is covered in the "Weaknesses" part.
- In equation (1), the formulation seems to be equivalent to taking the expectation with respect to the stochastic transition. This may potentially have some undesired consequences. Consider an example where we have that with probability (1-\epsilon), the next state s_{t+1}^i matches with the demonstration exactly, but with probability \epsilon, the next state deviates a lot from the demonstration, e.g. ||s_{t+1}^i - s_{t+1}^d||_2 -> infinity. According to equation (1), this demonstration would be really bad because, in expectation, the deviation is high (due to the large deviation with extremely small probability). Yet given the definition of feasibility defined in the paper, this demonstration is perfectly feasible (there exist actions which recover the demonstration with probability > 0, in fact with high probability). It would be great if the author can motivate this design choice of using the L-2 norm and potentially provide a few more choices of the reward function.
- It would be great if the authors can elaborate how the discount factor \gamma^f is chosen and what are the practical considerations regarding it.
- Does T contain all demonstrations or only a single demonstration?
- If there are two trajectories that happen to have the same state transition pair (s_t^d, s_{t+1}^d), but one is feasible and the other is not. Then what would w((s_t^d, s_{t+1}^d)) be in this case?
- What is the expected returns for the demonstrations in all the experiments? Are they higher than that of the proposed method or they are comparable?
- How is success rate defined for the Franka robot experiment?
Minor issues:
- Line 124: The authors mentioned that [7] requires "access to accurate inverse dynamics model". Although it seems like [7] proposes to learn the inverse dynamics model. The authors might want to clarify that in the later version.
- Line 156: The curly bracket should be closed after "\inT".
- In the appendices: what is a "disabled" panda robot?


**Reviewer Expertise:**

Very good: Comprehensive knowledge of the area

**Strengths And Weaknesses:**

Strengths:
- The paper considers an important problem of learning with demonstrations from potentially different dynamics. The problem itself is well-motivated: The introduction of the paper provides a concise but convincing example.
- The authors provide a comprehensive survey over related work, both broadly in the field of imitation learning, as well as the specific area of learning with demonstrations from different dynamics. The authors also pinpoint the difference between their new approach and existing work.
- The experimental evaluation is thorough. The author compare their proposed algorithm with state-of-the-art methods in the area and provide convincing results that the proposed method generally outperforms baselines in these tasks.

Weaknesses:
- The paper is overall technically sound. I like the f-MDP formation a lot since I think it is more suitable for stochastic MDPs than learning an inverse dynamics function, which is commonly used in the literature. However, there seems to be occasions where the authors use treatments that are not entirely appropriate for stochastic MDPs. For example, in line 113, the expectation should be taken w.r.t. the transition probability as well; in line 141, enforcing s_{t+1}^{I} = s_{t+1}^d might not be feasible for stochastic system, as it is sampled from a distribution; in line 187, "feasible trajectories still receive the maximal reward of 0" might not be true for stochastic MDPs based on the definition of feasibility from the authors. The authors are encouraged to go over these statements and potentially modify them in the next version of the manuscript.
- A potential weakness is that solving the f-MDP might require quite some environment interactions and computation. But it seems like the proposed method still outperforms existing methods given the same amount of interactions. But it would be great if the authors can provide more explanations on this. For example, how are the number of interactions for solving f-MDP and that for learning an inverse dynamics model compared to each other?

**Summary Of Recommendation:**

The paper proposed to solve an important and practical problem of learning with demonstrations from different dynamics. The paper is well-written and overall technically sound (with a few exceptions mentioned in the comments). The literature review is thorough and the novelty is clear. The experimental results are convincing.

---

> ### Author Response · Authors · 2021-08-25
> **Response1**
>
> We thank the reviewer for their constructive comments and feedback. We are excited that the reviewer thinks that our problem is well-motivated,  our introduction provides a concise and convincing example, and our evaluation is thorough. We would like to address the concerns brought up by the reviewer:
>
> **Stochastic MDP Formalism**
> We have modified and expanded on our stochastic MDP set up in Appendix A in the revision. The key point is to use the expected distance over all the possible next states given an action to measure the feasibility. Under such design, for the case where there is $\epsilon$ probability that the distance $||s_{t+1}^i - s_{t+1}^d||_2\rightarrow\infty $, the expected distance will also be $\infty$ and the feasibility will be 0. Such state transitions will be filtered out. Only those with a small expected distance will be preserved.
>
> **Different Metrics for the Reward Function**
> We use the L2 distance in our method description and the experiments because, in all our environments including the Mujoco and Robot environments, the L2 distance is an appropriate metric for measuring the distance between states. However, the reward in our f-MDP is not restricted to the L2 distance and can be changed depending on the specific choice of state space. For example, for a state space with unit vectors, we can use the cosine distance as the distance metric.
>
> In Fig. 4 in the Appendix of the revision, we show the expected return of our method by using different distance metrics in the Swimmer environment. We use L1 distance and Cosine distance (the cosine of the angles between two state vectors) as examples. We observe that L1 distance performs close to L2 distance, but Cosine distance performs worse than L2 distance because Cosine distance only cares about the distance on angle but ignores the scale of the vectors, while in Swimmer, the scale of the states is also important. The results show that the choice of this distance function is flexible and depends on the specific state space used in the problem setting.
>
> **Comparison between the Number of Interactions Required for Learning an f-MDP vs an Inverse Dynamics Model**
> Our goal is to learn the feasibility of the demonstrations, and the most efficient approach is to only focus on learning the feasibility and nothing more. Learning the inverse dynamics model typically needs interaction data covering the whole state-action space instead of the state-action subspace contained in the demonstrations. Our f-MDP is a simple MDP with a discrete initial state distribution defined by the initial states of the demonstrations. We use RL in a more effective manner to explore the state space when starting with this finite initial distribution provided by the demonstrations. In addition, our reward function encourages the policy to explore near the state-action subspace of the demonstrations. Therefore, learning a policy using our f-MDP is more efficient than relying on the exploration of all of the state-action space as it is often needed for learning an inverse dynamics model.
>
> Furthermore, the f-MDP enables learning an accurate policy, which can estimate the feasibility very accurately. However, learning the inverse dynamics model for a continuous state-action space with finite interactions may not lead to learning an accurate model and instead lead to errors that can be accumulated and exaggerated in a sequence when estimating the feasibility for a trajectory.
>
> In our experiments, we use the same number of interactions for learning the f-MDP and learning the inverse dynamics model in ID-Feas for a fair comparison of their performance. This is shown in Fig. 3. As an example, we use 5e5 interaction steps to learn our feasibility policy and the inverse dynamics model for ID-Feas in the Swimmer environment in Fig. 3. Our method leads to expected return of 74.89$\pm$19.68 for and ID-Feas learns to expected return 48.96$\pm$38.50. We observe that our f-MDP learns the feasibility more efficiently.
>
> **Discount Factor $\gamma$**
> The discount factor $\gamma^f$ depends on how fast the compounding error increases with respect to the number of steps in the environment. Faster increasing compounding errors need smaller $\gamma^f$. In practice, this depends on the scale of the ‘movement’ of the agent at each step. For example, for a robot arm, if each joint can only move a small angle at each time step, we can set $\gamma^f$ to be larger or if each joint can move a large amount at each time step, $\gamma^f$ should be set to a smaller value. In our experiments, we fix $\gamma^f=0.9$, which works well empirically for all of our environments. We have clarified this in the Appendix in the revision.
>
> **Set $T$**
> $T$ contains state transitions in all of the demonstrations.

---

> > ### Author Response · Authors · 2021-08-25
> > **Response2**
> >
> > **State Transition Appearing in Multiple Trajectories with Different Feasibility**
> > For a state transition appearing in multiple trajectories, we use the sum of the feasibility of all the trajectories it belongs to. This way, if a state transition only appears in infeasible trajectories---even when appearing multiple times---the state transition will still have low feasibility. If a state transition appears at least in a feasible or nearly feasible trajectory, the feasibility value will be high because the state transition is part of a useful demonstration.
> >
> > **Expected Return of the Demonstrations**
> > The demonstrations are all optimal demonstrations for different demonstrators but they may not be feasible for the imitator. Here we provide the average expected return of the demonstrators for the different environments:
> >
> > ||Swimmer | Walker2d (first)| Walker2d (first) | HalfCheetah (first) |  HalfCheetah (second) |  Simulated Robot | Real Robot |
> > |:---:|:---:|:---:|:---:|:---:|:---:|:---:|:---:|
> > |Average Expected Return| 106$\pm$3 | 3098$\pm$118 | 3720$\pm$336 | 3229$\pm$170 | 3337$\pm$67 | 1823$\pm$110 | 2531$\pm$ 362|
> > |Performance of Ours | 75$\pm$20 | 3147$\pm$10|3424$\pm$645 |2832$\pm$291 |3142$\pm$89|2127$\pm$5053|2746$\pm$2712 |
> >
> > We observe that in the first setting of the Walker2d environment, the simulated robot, and the real robot environments, our approach performs comparably to the expected return of the optimal demonstrations. In the Swimmer, the second setting of Walker2d, and the HalfCheetah environments, the performance is worse than the expected return of the optimal demonstrations. This is because only a few demonstrations are feasible for the imitator and that may not be enough to learn an optimal policy. However, the margin between our approach and the demonstrations is still not very large. The results show that our proposed feasibility can select useful demonstrations for the imitator to imitate.
> >
> > **Definition of Success Rate**
> > In the Franka robot experiment, success is defined as when the robot arm moves the book to the empty area of the shelf without colliding with the other books or the shelf within the time limit. The success rate is the average of successful runs over 100 rollouts with different initial states.
> >
> > **Minor Comments**
> > Thanks for pointing these out. Ref. [7] learns the inverse dynamics model but the accuracy of the feasibility highly relies on the accuracy of the inverse dynamics model. Also, the availability of such an inverse dynamics model is included as a limitation of their paper. We added this explanation to the revision in Sections 1 and 3.
> > In addition, we revised the bracket typo in the revision. The ‘disabled’ panda robot is the 3-DoF robot arm and the ‘non-disabled’ panda robot is the 7-DoF robot arm in our Franka Panda environment.

---

> > > ### Comment · Reviewer_Srqj · 2021-09-03
> > > **Thank you for your response**
> > >
> > > The authors have addressed most of my concerns. In particular, I really like that the authors provide additional comparison with different distance metrics in Appendix D.5, the discussion on the choice of discount factor $\gamma^f$ in Appendix C.3, and report the performance of the experts in Table 3 in the appendix. However, some of treatments of deterministic versus stochastic dynamics is still not fully corrected. For example, in line 114, the expectation should be taken with respect to the stochastic dynamics as well. The authors also use $p$ to denote both probability (line 104-105, line 135) and the transition function (e.g. Figure 2, line 145, etc.), which could be confusing from time to time. I hope that the authors can double check these in the later version. In the experiment section, the authors might also want to highlight that they "use the same number of interactions for learning the f-MDP and learning the inverse dynamics model in ID-Feas for a fair comparison of their performance".
> > >
> > > With that being said, I think this paper is a solid paper. It considers an important research problem and proposes a practical approach to it. The overall presentation of the paper is very clear and I enjoyed reading it. The additional results provided during the discussion period also significantly strengthened the paper. I recommend this paper being accepted to CoRL. (Though I will keep my score as it is since I am not entirely sure what does "strong" acceptance mean...I would definitely increase my score if there were a score between weak and strong acceptance)

---

> > > > ### Author Response · Authors · 2021-09-03
> > > > **Response**
> > > >
> > > > Thank you for your suggestions. We will make sure to address these final edits in the final version of the paper.

---

### Official Review · Reviewer_fG77 · 2021-07-22

**Originality:** Good
**Technical Quality:** Good
**Clarity Of Presentation:** Very Good
**Impact:** 3

**Recommendation:**

Weak Accept: I recommend accepting the paper, but will not argue for my recommendation if the majority of other reviewers have a different opinion.

**Summary:**

This paper attempts to improve the algorithm from the perspective of Feasibility of Imitate Learning, and encourage the imitator to learn from more informative demonstrations by defining a new reward for the learning task of Feasibility. In addition, the theory in the article is preliminarily verified through simulation experiments and real robotic arm experiments. The results are basically credible and have some positive effects on the development of this field.

**Issues:**

1. It is not reasonable enough that this paper uses robotic arms with different degrees of freedom to cite its ideas. In robotics, if the robot arm can be manipulated to complete the RL task, the dynamic model of the robot arm is almost known, at least it is not so difficult to obtain the dynamic model.
2.In Figure 1, it should use the robot arm picture consistent with the experiment, instead of the red robot arm in the picture (it can use the Mark-On-Image method to emphasize 3-DOF or 7-DOF).
3.Line 134-135, the statement about perfect inverse dynamics model is not clear enough.
4.Line 179，if the Eqn.2 is not used here, the new formula should be mentioned in the text as soon as possible. For readers, the final method used in this idea is very important.
5.Line 185, for the traditional RL reward function, the use of gamma is the basis of RL theory, so the statements in the article should be reduced as much as possible.


**Reviewer Expertise:**

Very good: Comprehensive knowledge of the area

**Strengths And Weaknesses:**

This paper attempts to introduce the concept of feasibility Markov Decision Process (f-MDP), and  to find a solution to the problem of Imitate Learning with Different Dynamics from a higher perspective. This is beneficial for obtaining new ideas. However, in the process of exploring feasibility score or metric, the use of norm and parameter gamma is extremely similar to the traditional RL method. What’s more, this paper uses distance as a feasibility evaluation metric, which may in fact do not sufficiently reflect feasibility, even it seems more like following the demonstration trajectory. Actually, the f-MDP is very similar to ordinary MDP, and there is little obvious difference except for the description.

**Summary Of Recommendation:**

It is recommended to adjust the idea and experimental plan to make the concept of feasibility more prominent and reasonable.

---

> ### Author Response · Authors · 2021-08-25
> **Response**
>
> We thank the reviewer for their constructive comments and feedback. We are excited that the reviewer liked the idea of f-MDP and consider it beneficial for learning from suboptimal demonstrations. We would like to address the concerns brought up by the reviewer:
>
> **Similarities to Traditional RL**
> We would like to emphasize that one of our key contributions in this paper is the introduction of the f-MDP for the goal of assessing the feasibility of trajectories.  However, we never intended to propose a new reinforcement learning algorithm for solving the f-MDP. In fact, we do use prior RL algorithms to learn a policy for the f-MDP.
>
> We also would like to emphasize there is a different underlying reason for using the discount factor $\gamma^f$ in our setting. Specifically the trajectory distance in future steps can deviate largely due to the compounding errors and is potentially not discriminative enough to distinguish nearly feasible and completely infeasible trajectories. So we define our reward as the state distance between the demonstrations and rollouts from the policy, and hence we discount the distance in future steps using $\gamma^f$ on the reward.
>
>
> **L2 Distance for Measuring Feasibility**
> We use the L2 distance in our method description and the experiments because in all our environments including the Mujoco and Robot environments, the L2 distance is an appropriate metric for measuring  the distance between states. However, the reward in our f-MDP is not restricted to the L2 distance and can be changed depending on the specific choice of the state space. For example, for a state space with unit vectors, we can use the cosine distance as the distance metric.
>
> In Fig. 4 in the Appendix of the revision, we show the expected return of our method by using different distance metrics  in the Swimmer environment. We use L1 distance and Cosine distance (the cosine of the angles between two state vectors) as examples. We observe that L1 distance performs close to L2 distance, but Cosine distance performs worse than L2 distance because Cosine distance only cares about the distance on angle but ignores the scale of the vectors, while in Swimmer, the scale of the states is also important. The results show that the choice of this distance function is flexible and depends on the specific state space used in the problem setting.
>
>
> **Similarities between the f-MDP and Ordinary MDP**
> The f-MDP is also an MDP and thus the components of f-MDP are similar to an ordinary MDP including state space, action space, transition function, reward function, etc. However, we want to clarify that the design of each component of our f-MDP is different from an ordinary MDP and is designed particularly for learning the feasibility. As shown in the Section 4 (‘One-step f-MDP’ and ‘Trajectory f-MDP’), the f-MDP is designed based on the original ordinary MDP representing the problem setting as well as  the demonstrations. The initial state space is designed as the initial states of the demonstrations while the reward is designed as the per-step state distance. These key designs differentiate f-MDP from an ordinary MDP and enable f-MDP to learn the feasibility.
>
>
> **Known Dynamics Model for Robot Arm Experiments**
> Not having access to the dynamics model of an agent or robot is a fundamental problem in robotics and there is a plethora of work focused on learning inverse dynamics models [a,b]. Even though we may know the dynamics model of the Franka robot arm, here we use the robot arm experiments and vary the degrees of freedom of the robot as a proof of concept to demonstrate our feasibility model on a real robotics system outside of simple simulations. This experiment also  supports  a realistic situation where we want to learn a policy for a robot but we only have demonstrations from other robots (different hardware or number of joints as we show in our experiments) with the same state space.
>
> **Other Minor Issues**
> For issues 2,3,4, we have revised the paper accordingly. For issue 4, the organization of Eqn. (2) and the final feasibility score in Eqn. (3) is because we use the discount factor in Eqn. (3) and use the trajectory feasibility for all the state transitions in a trajectory. We need to first explain these reasons and then the final feasibility equation can be derived. We have re-organized the text in Section 4 to make it more clear.
>
> [a] Agrawal, Pulkit, et al. "Learning to poke by poking: experiential learning of intuitive physics." Proceedings of the 30th International Conference on Neural Information Processing Systems. 2016.
>
> [b] Hafner, Danijar, et al. “Learning Latent Dynamics for Planning from Pixels.” Proceedings of the 36th International Conference on Machine Learning, 2019.

---

> > ### Comment · Reviewer_fG77 · 2021-09-02
> > **Reviewer Response**
> >
> > In general, the authors have a clear explanation of f-MDP and theoretical details.
> >
> > The concept of Learning Feasibility in the latest version of the paper has positive academic significance for coRL, and I believe that it also has some contributions to RL theory.

---

### Official Review · Reviewer_GBdL · 2021-07-23

**Originality:** Very Good
**Technical Quality:** Excellent
**Clarity Of Presentation:** Excellent
**Impact:** 4

**Recommendation:**

Strong Accept: I recommend accepting the paper and will argue for my recommendation even if other reviewers hold a different opinion.

**Summary:**

The paper is concerned with learning movement primitives from demonstrations collected on robots with dynamics that are different than that of the imitator robot (similar to the correspondence problem). Specifically, the paper contributes a method to compute a metric that quantifies the feasibility of the imitator robot reproducing a given demonstration. The available demonstrations are then weighted based on this feasibility metric so that the imitator can prioritize demonstrations are more suitable for the imitator's dynamics. Experimental results (both in simulation a real robot platform) demonstrate the utility of the proposed approach and its and relative benefits over existing approaches.

**Issues:**

1. How do we know that the assumption "by imitating useful demonstrations, we can learn the optimal policy" is valid for a given setting? How strong is it? What are the consequences of the assumption being false?

2. In the introduction, please provide a brief and conceptual explanation of what it means to assume access to the "environment" of the imitator. For instance, state how the environment is modeled, and what is assumed to be given.

3. What if the assumption that the reproductions that initially diverge are going to diverge more in the later time step is wrong. For instance, what if the "closest" or "best" reproduction that the imitator could produce requires the imitator to deviate from the demonstration in the very initial time steps of the demonstrations and then converge back close to the demonstrations? In other words, how valid is the assumption that a discount factor is needed?

4. On a related note, what if parts of the demonstrations are useful/feasible, while other parts are harmful/infeasible? Could the score be adapted to address this?

5. The narrative in the paper makes it seem like Ref. [7] just assumes an accurate inverse dynamics model. But, from a quick glance, it appears that Ref. [7] in fact learns the inverse dynamics model. If this is indeed the case, please clarify in the manuscript it avoid unintentionally misleading the readers.

**Reviewer Expertise:**

Very good: Comprehensive knowledge of the area

**Strengths And Weaknesses:**

**Strengths**

+ The work is well motivated by the need to leverage already available demonstrations, especially when they are collected on a robot with a different morphology/dynamics.
+ The paper is well written and was a joy to read.
+ The contributions are sufficiently contextualized within the existing literature on imitation learning.
+ The method does not assume knowledge of the exact inverse dynamics model of the imitator, and relies only on knowledge of the imitator's environment.
+ By moving away from a binary feasibility function, the proposed method does not completely discard infeasible trajectories that could still contain useful information.
+ The figures in the paper are well designed and composed. They prove to be very helpful illustrations of various core concepts in the paper.
+ The probability distribution computed in Eq. (4) that models the usefulness of a demonstrators is very helpful from a practical stand point as we can focus our energy on collecting additional demonstrations from more useful platforms if necessary.
+ The experiments are well designed and conclusively demonstrate the utility of the proposed approach. Further, careful comparison with SOTA approaches illustrate the relative benefits that the proposed approach brings to the table. Further, the analysis in Section 5.3 is illuminating and commendable.

**Weaknesses**

- As the paper acknowledges, the idea of learning a feasibility score is not novel and has been proposed in Ref. [7]. Having said that, this is a minor weakness for a couple of reasons. First, it is important to note that Ref. [7] was published only in 2021. Second, the paper explicitly includes Ref. [7] as a baseline in the experiment and demonstrates the superiority of the proposed method.
- A key assumption made by this work is that "by imitating useful demonstrations, we can learn the optimal policy". While this assumption is a direct result of the no-free-lunch theorem given the problem setup, it is currently buried at the end of Section 3. This assumption should be acknowledged more prominently and its consequences explicitly discussed.
- While the experiments compare the performance of the proposed approach against some existing imitation learning algorithms, it does not compare against approaches that map demonstrations to the imitator. Granted that the demonstrations might not be feasible and the mapping might produce demonstrations are far from the intended demonstrations. As the paper claims, such mapping "may lead to unknown behavior".  But, it would be very helpfully experimentally verify this claim and demonstrate that the proposed method (feasibility-based weighting) indeed performs better than a policy learned from mapped (and potentially problematic) demonstrations.
- The methods assumes that a single number uniformly defines the feasibility of a demonstrated trajectory. What if parts of the demonstrations are useful/feasible, while other parts are harmful/infeasible?

**Summary Of Recommendation:**

This paper presents a much-needed approach to handling demonstrations that are collected from a heterogeneous set of robot platforms that have potentially different dynamics. Importantly, the approach leverages information from such demonstrations without needed the exact inverse dynamics model. Save for some minor weaknesses, this is a solid conference paper that presents an approach that will likely have a broad impact. While the paper is mostly focused on learning primitives for manipulators. The key ideas are much broader and can be extended to various applications, such as autonomous vehicles and multi-agent systems.

---

> ### Author Response · Authors · 2021-08-25
> **Response**
>
> We thank the reviewer for their constructive comments and feedback. We are excited that the reviewer thought our problem is well-motivated, the paper is well-written, the experiments are well-designed, and conclusively demonstrate the utility of our approach. We would like to address the concerns brought up by the reviewer:
>
> **Validity of our Assumption:‘by imitating useful demonstrations, we can learn the optimal policy’**
> The assumption requires that there are sufficient demonstrations for an imitation learning algorithm to learn an optimal policy. We emphasize that this is a general assumption for imitation learning but not a particularly strong assumption for our work. The assumption is also adopted in standard imitation learning works [2,4,8] and imitation learning from different dynamics works [5,7]. Note that our work focuses on developing the feasibility score to detect all the useful demonstrations and then uses standard imitation learning algorithms to learn from these useful demonstrations, so we inherit this assumption from prior imitation learning works. If the assumption is violated, as shown in these prior works, a suboptimal policy will be learned. We have made this assumption more prominent and have further explained it in the revision.
>
> **Experiments with Approaches that Map Demonstrations to the Imitator**
> The state-of-the-art demonstration mapping method (DCC) [26] requires random trajectories from both the demonstrator and the imitator. However, we only have access to demonstrations but not demonstrators’ environments. We use demonstrations and the imitator’s random trajectories to experiment with this demonstration mapping method. Due to the time limit, we have only conducted the experiments in the Swimmer environment. In the Table below, we observe that the performance of DCC is much worse than our method and even worse than GAIL. This is because the demonstrations and the imitator’s random trajectories are not aligned and the mapping method usually needs many trajectories to learn a generalizable mapping while here the number of demonstrations is too small to support learning such a mapping. We have added these results in Appendix D.6.
>
> |Method|GAIL|SAIL|DCC|ID-Feas|Ours|
> |:---:|:---:|:---:|:---:|:---:|:---:|
> |Expected Return|31.20$\pm$22.25|0.56$\pm$4.27|5.32$\pm$3.43|48.96$\pm$38.50|74.89$\pm$19.68|
>
> **What if Parts of the Demonstrations are Feasible**
> This is an interesting point brought up by the reviewer. Currently, standard imitation learning algorithms as well as our algorithm rely on learning from the full trajectory from the start to the end state. If a segment of the trajectory is far from feasible or harmful, then the remaining part is also not going to be useful for our algorithm. Therefore, we only learn from trajectories that are helpful in all parts.
>
> However, one can extend imitation learning algorithms including ours by first segmenting the trajectory into helpful and harmful segments, and learn from these segments. However, there need to be two core assumptions in place: 1) there exist useful partial trajectories that can connect other useful segments (i.e, there is coverage by useful trajectories over all states in the demonstrations); 2) the useful partial trajectories are feasible under the transition function.
> This is outside of the scope of this work, but an interesting potential extension.
>
> **Discount Factor**
> If a trajectory is not feasible in the initial time steps but converges back to be feasible in the later steps, this trajectory is still not useful since there is no feasible partial trajectory to lead the agent from the initial state to the later feasible parts of the trajectory. As we discussed in the last paragraph, we only learn from trajectories that are helpful in all parts. The later steps are not important in deciding the feasibility of the trajectory and discounting them does not influence the final imitation learning performance much. So the discount factor can still be used in this situation.
>
> **Ref. [7] Learns the Inverse Dynamics Model**
> Thanks for pointing this out. Ref. [7] learns the inverse dynamics model. However, the accuracy of the feasibility highly relies on the accuracy of the learned inverse dynamics model. Also, the availability of such an inverse dynamics model was mentioned as a limitation of their paper. We have added this explanation to the revision in Sections 1 and 3.
>
> **Explanation of Access to the ‘environment’**
> In the revision in Section 1, we state that the imitator is modeled by an MDP. We discuss the details and components of this MDP which represents our environment in Section 3.

---

### Meta-Review · Area_Chair_Yf2y · 2021-08-13

**Recommendation:** Accept (Poster)
**Confidence:** 4

**Metareview:**

This paper presents an approach for learning from demonstrations that allows the imitator to distinguish demonstrations feasible for their specific action spaces and the slight difference between the dynamics of demonstrators and imitators. Some strong statements are not supported by the experimental results, e.g. "we would like to be able to leverage demonstrations collected from agents with different dynamics." In the HalfCheetah, Swimmer examples the actions space of the imitator and demonstrator are different whereas in the Walker2d the friction coefficient between the demonstration and mutation scenarios are different.
The idea of identifying the demonstrations for a specific kinematic and dynamic chain of imitator is very relevant CoRL scope. Reviewers raised some concerns regarding the presentation of the paper and the corresponding concepts. For instance, there are similarities between the proposed approach and traditional RL. Moreover, the L2-norm distance between the demonstration and imitation trajectories may not be always useful in robotic tasks: (1) the trajectories may not be aligned temporally — which is not discussed in this paper; and (2) the L2 norm distance may not sufficiently reflect the feasibility (a) trajectories either are expressed in task-space or (b) the trajectories are expressed in join space, hence the L-2 norm distances for orientation and position or the distance between 3 DOF and 7 DOF must be clarified. A clear description of the problem definition and problem formulation can improve the quality of the paper.

The limitation of the approach must be clearly discussed. It should be clarified to what extent the approach can deal with different dynamics: (1) is it only reduced DOF of a manipulator the change in the dynamic, (2) or the change of friction coefficient in the interactive tasks; or (3) the change of dynamic model, e.g. the Mass, Coriolis and Gravity Matrices are different.


**Quality**: Needs improvement. Some claims are not fully supported by the results which can be improved. E.g. while the demonstrator and imitator manipulators in the motivating example of Fig. 1 have different kinematic chains, the experimental results only show a Franka arm as the demonstrator (with 3 and 7 DOFs) and imitator (with 3 DOF) where their kinematic chain is the same. Some additional experiments or analyses are needed to support the claims about the use of the approach for different dynamics— this term needs a clear definition at the beginning of the paper.

**Clarity**: The paper is clear after addressing the concerns raised by the reviewers.


**Significance**: The paper has practical significance and the theoretical significance is clear after response to reviewers.

**Originality**: The author presents a novel idea which computes the feasibility metric for a kinematic/dynamic chain and a demonstration. Nonetheless, the authors need to describe what is the similarity and differences between the proposed approach and traditional RL.

---

> ### Author Response · Authors · 2021-08-25
> **Response**
>
> We thank the reviewers for their constructive feedback and the meta-reviewer for the insightful suggestions. We appreciate that the reviewers agree that the problem setting of imitating demonstrators with different dynamics is interesting and the paper proposes a novel method for learning feasibility. The reviewers mainly have questions on the difference between f-MDP and ordinary MDP, the feasibility under stochastic MDPs, and the clarification of feasibility. We have 1) emphasized the contribution of designing the f-MDP to learn feasibility and have clarified the difference between f-MDP and the regular MDP that formalizes our problem; 2) expanded on how our method works in the stochastic MDP case; 3) clarified other minor concerns about the feasibility such as the choice of distance metric. In the revision, we use text in blue color to indicate the text we have revised.
>
> We also have addressed the points brought up by the meta-reviewer here:
>
> **Problem Definition and the Term 'Dynamics'**
> Our problem setting targets learning a policy from demonstrations that are collected from different demonstrators with different dynamics. Here, the term ‘dynamics’ refers to the ‘transition function $p$’ in the MDP, which defines the probability of the next state given the current state and action. This ‘transition function’ can be changed with respect to the change of many factors such as the action space, the degrees of freedom, or the friction coefficient. We apologize if this definition was unclear and have clarified this in the revision in Sections 1 and 3.
>
> **Statements are not Supported by Experimental Results**
> As mentioned above, we use the term ‘different dynamics’ to refer to differences in the state transition function of the MDP.  Under this definition, our claim that ‘we would like to be able to leverage demonstrations collected from agents with different dynamics.’ is supported by our experimental results, as we make changes in the joint limits, friction, and control force in the Mujoco environment as well as changes in the degrees of freedom in the Franka Panda robot arm environment. These changes lead to varying transition functions (or ‘dynamics’ as we use in our paper).
>
> We have added new experiments in the Mujoco Hopper environment with different gravity matrices by varying the gravitational constant. We show the exact gravitational constants of different demonstrators in Appendix C.1 and the results in Appendix D.1 in the revision. We demonstrate that our approach outperforms baseline methods under gravity changes.
>
> **Difference between Our Method and Traditional RL**
> As we explain in our response to Reviewer fG77, we emphasize that our contribution focuses on proposing the f-MDP for the goal of learning the feasibility of trajectories from demonstrators with different dynamics (transition functions). Our key insights are designing the components of the f-MDP to accommodate learning an accurate feasibility measure. After designing the f-MDP, the following step is to learn a policy in f-MDP. The f-MDP formulation and the following policy learning are theoretically de-coupled. To learn the policy from f-MDP, we actually use prior RL algorithms.
>
> **Differences between f-MDP and Ordinary MDP**
> There are several differences between f-MDP and ordinary MDP. We have further explained this in our response to Reviewer fG77.
>
> 1) The goal of f-MDP is to learn the feasibility for a particular set of demonstrations while the goal of an ordinary MDP is to learn an optimal policy for the task defined by the MDP. For example, we use a regular MDP to formalize our problem. Then we define the f-MDP and find the optimal policy of the f-MDP to assess the feasibility of the demonstrations. Using the feasibility learned from the optimal policy of f-MDP, we use standard imitation learning techniques to learn the optimal policy of our original regular MDP.
>
> 2) The ordinary MDP is independently designed for a task by the task designer, while the f-MDP is defined based on the demonstrations and the original ordinary MDP formalizing the problem. The initial state space is designed as the initial states of the demonstrations while the reward is designed as the per-step state distance.
>
> 3) Though both f-MDP and ordinary MDP use the discount factor $\gamma$, the underlying reasons for using the discount factor are different. Using the discount factor in ordinary MDP is because the immediate reward is more important than the reward in the future while using the discount factor in f-MDP is because the trajectory distance in future steps can deviate largely due to the compounding errors and is not discriminative enough to divide nearly feasible and completely infeasible trajectory.
>
> These key designs differentiate f-MDP from ordinary MDP and demonstrate that the f-MDP is a non-trivial design to accommodate learning the feasibility, and enabling standard imitation learning techniques to be used even with suboptimal demonstrations.

---

### Decision · Program_Chairs · 2021-09-13

**Decision:**

Accept (Poster)

**Comment:**

This paper presents an approach for learning from demonstrations that allows the imitator to distinguish demonstrations feasible for their specific action spaces and the slight difference between the dynamics of demonstrators and imitators. Some strong statements are not supported by the experimental results, e.g. "we would like to be able to leverage demonstrations collected from agents with different dynamics." In the HalfCheetah, Swimmer examples the actions space of the imitator and demonstrator are different whereas in the Walker2d the friction coefficient between the demonstration and mutation scenarios are different.
The idea of identifying the demonstrations for a specific kinematic and dynamic chain of imitator is very relevant CoRL scope. Reviewers raised some concerns regarding the presentation of the paper and the corresponding concepts. For instance, there are similarities between the proposed approach and traditional RL. Moreover, the L2-norm distance between the demonstration and imitation trajectories may not be always useful in robotic tasks: (1) the trajectories may not be aligned temporally — which is not discussed in this paper; and (2) the L2 norm distance may not sufficiently reflect the feasibility (a) trajectories either are expressed in task-space or (b) the trajectories are expressed in join space, hence the L-2 norm distances for orientation and position or the distance between 3 DOF and 7 DOF must be clarified. A clear description of the problem definition and problem formulation can improve the quality of the paper.

The limitation of the approach must be clearly discussed. It should be clarified to what extent the approach can deal with different dynamics: (1) is it only reduced DOF of a manipulator the change in the dynamic, (2) or the change of friction coefficient in the interactive tasks; or (3) the change of dynamic model, e.g. the Mass, Coriolis and Gravity Matrices are different.


**Quality**: Needs improvement. Some claims are not fully supported by the results which can be improved. E.g. while the demonstrator and imitator manipulators in the motivating example of Fig. 1 have different kinematic chains, the experimental results only show a Franka arm as the demonstrator (with 3 and 7 DOFs) and imitator (with 3 DOF) where their kinematic chain is the same. Some additional experiments or analyses are needed to support the claims about the use of the approach for different dynamics— this term needs a clear definition at the beginning of the paper.

**Clarity**: The paper is clear after addressing the concerns raised by the reviewers.


**Significance**: The paper has practical significance and the theoretical significance is clear after response to reviewers.

**Originality**: The author presents a novel idea which computes the feasibility metric for a kinematic/dynamic chain and a demonstration. Nonetheless, the authors need to describe what is the similarity and differences between the proposed approach and traditional RL.